# Resect and Retrieve Colorectal Polyps: Time for New Insights

**DOI:** 10.3390/jcm14165846

**Published:** 2025-08-18

**Authors:** Giulia Gibiino, Cecilia Binda, Matteo Secco, Lorenzo Cosentino, Francesco Poggioli, Stefania Cappetta, Davide Trama, Andrea Fabbri, Chiara Coluccio, Carlo Fabbri

**Affiliations:** 1Gastroenterology and Endoscopy Unit, Forlì-Cesena Hospitals, AUSL Romagna, 47121 Forlì, Italy; cecilia.binda@gmail.com (C.B.); lorenzo.cosentino@auslromagna.it (L.C.); davide.trama@auslromagna.it (D.T.); andrea.fabbri2@auslromagna.it (A.F.); colucciochiara@gmail.com (C.C.); carlo.fabbri@auslromagna.it (C.F.); 2S. C. of Gastroenterology and Digestive Endoscopy, Michele e Pietro Ferrero Hospital, 12060 Verduno, Italy; matteo.secco01@gmail.com; 3Department of Medical and Surgical Science, University of Bologna, 40138 Bologna, Italycappetta.stefania@gmail.com (S.C.)

**Keywords:** colorectal adenoma, endoscopic resection, surveillance colonoscopy

## Abstract

Polyp retrieval following colorectal polypectomy remains a critical step for histopathological analysis and determining appropriate surveillance intervals. Despite reported retrieval rates exceeding 90% in the literature, significant heterogeneity persists in clinical practice, particularly for polyps < 10 mm, due to the lack of standardized retrieval methods. This review synthesizes current evidence on polyp retrieval techniques, the impact of lesion size, and device-specific considerations, including suction-based methods, retrieval nets, and other approaches such as the water-bolus and water-slider techniques. We also examine the clinical utility and limitations of the “resect and discard” and “diagnose and leave in situ” strategies, highlighting barriers to widespread implementation such as medico-legal risks, variability in optical diagnosis, and discrepancies across international guidelines. The integration of advanced imaging technologies, including high-definition endoscopy, virtual chromoendoscopy, and artificial intelligence-driven computer-aided diagnosis (CADx), represent promising tools to help in increasing the diagnostic accuracy of diminutive polyps. As post-polypectomy surveillance recommendations remain tethered to histological findings, this review underlines the need for updated, evidence-based frameworks that take into account technological advancements while ensuring diagnostic precision, cost-effectiveness, and patient safety in colorectal cancer prevention.

## 1. Introduction

Polyp retrieval after a polypectomy is crucial for histological evaluation and to determine the optimal subsequent follow-up interval [1]. Although not much data are available on global polyp retrieval rates, studies from the last ten years report a 92–93.9% retrieval in colonoscopies [2,3].

Furthermore, polyps resection and retrieval have been a hot topic in the endoscopist’s community in the last decade since the advent of “resect and discard” and “diagnose and leave in situ” approaches.

The resect and discard (RD) approach is essentially based on the polyp location and dimension and can be resumed as everything that is diminutive proximal to the sigmoid is considered an adenoma, and everything diminutive in the rectosigmoid is hyperplastic [4]. Even so, it must be highlighted that “resect and discard” is barely implemented in the USA and Europe, even in academic centers, where optical diagnosis has resulted in surveillance interval assignments that exceed the thresholds set by ASGE and ESGE [5]. In the literature, the risk of invasive cancer in lesions ≤ 5 mm is reported as low as 0.009%, making this event negligible; meanwhile, most interval cancers are the result of a missed lesion in a previous colonoscopy [6]. Even so, the slow adoption of RD in common clinical practice is mostly due to the perceived medicolegal risk of patients claiming that cancer arose in a diminutive lesion that was thrown away, leading to incomplete treatment. Moreover, adoption of the RD strategy without understanding its scientific rationale could force endoscopists to suggest a follow-up colonoscopy at the shorter end of the recommended interval, deleting the economic and clinical advantage of this approach.

In the “diagnose and leave in situ” approach, the endoscopist is allowed not to resect diminutive polyps if optical diagnosis using high-definition white light instruments and magnification techniques, such as Narrow Band Imaging (NBI) or Blue Light Imaging (BLI), can grant an accurate prediction of polyp histology. This approach is obviously faster than the resect and discard and less expensive but leaving the benign lesion in situ poses the same problems as resect and discard to the endoscopist. The real game changer for this strategy is represented by the advent of artificial intelligence, where computer-aided diagnosis (CADx) has shown great results, making optical diagnosis safe and easily available for all endoscopists and patients. Hassan and all recently showed that CADx exceeded the benchmarks required for the optical diagnosis of colorectal polyps, making diagnosis and leave in situ a possibility for every endoscopist using CADx, no matter the expertise [7].

The changing strategies of polypectomy and retrieval, from diminutive or small polyps to advanced resections, require an updated reflection on this issue.

## 2. Methods

We selected articles discussing polypectomy techniques and retrieval methods. We developed a non-systematic review article using the following electronic sources: PubMed, EMBASE, Google Scholar, Ovid, MEDLINE, Scopus, the Cochrane controlled trials register, and Web of Science. We used the following search terms singly and in combination: “Polyp resection”, “Polypectomy”, “polyps and leave-in situ strategy”, “polyp retrieval”, “specimen retrieval”, “snare polypectomy and retrieval”, “polyp suction”, “polyp roth net retrieval”, and “resect and discard strategy”. We examined all the articles reporting data related to humans (inclusion criterion), while excluding works with no full text available, works that were not in the English language, book chapters, and abstracts (exclusion criteria). Finally, we evaluated supplementary references among the articles evaluated in the first search round.

## 3. Results

### 3.1. Polyps Retrieval in Current Practice—Techniques and Devices

Successful polyp retrieval may depend on several factors including resection techniques, lesion size, and specimen collection. Practice is mainly influenced by the experience of the individual center, and there are no specific recommendations or agreements on how to retrieve colorectal polyps, especially if they are <10 mm, in a standardized manner. We discuss these in detail below.

#### 3.1.1. Resection Techniques

Cold forceps polypectomy (CFP) has two major advantages: complete retrieval—0% in retrieval failure vs. 5% after cold snare polypectomy for diminutive polyps (dimension ≤ 5 mm) [8]—and the ability to continue endoscopic examination without withdrawing the endoscope; however, resection can be incomplete and limited to diminutive polyps ≤ 3 mm.

Snare polypectomy—both hot snare polypectomy (HSP) and cold snare polypectomy (CSP)—has lower retrieval rates (93–100%) [9] but ensures a higher rate of complete resection, especially for polyps > 3 mm [8]; for lesions smaller than 10 mm, polyp retrieval rates do not seem to differ between CSP and HSP (97% for both) [1], although it is unclear which technique is superior in terms of complete resection [1,9,10,11]. The thickness of the snare in CSP does not seem to influence the rate of retrieval, though the “polyp flying away” phenomenon immediately after resection seems more common [12,13]. In such cases, if the excised polyp migrates after resection, it is useful to spray water through the accessory channel and follow the path of the stream in order to search for the resected specimen, which is likely floating in the first proximal or distal pool of water the endoscopist encounters (water-bolus method) [14]. Some authors report that underwater CSP vs. conventional CSP guarantees easier and quicker retrieval and higher retrieval rates with no risk of polyp fragmentation [15].

#### 3.1.2. Size of the Polyp

The risk of loss of resected tissue is higher for polyps ≤ 5 mm [3,16]. If the polyp or fragment can be moved through the instrument channel, which is generally 2.8 to 3.7 mm in size, it can be placed in the 5 or 6 o’clock position and suctioned (unless it is directly removed by forceps) [17]; polyps larger than the channel size up to 10 mm can also be suctioned, though with the risk of polyp deformation [14]. In these cases, suctioning can be performed by removing the suction valve on the instrument and occluding the open suction valve with the index finger, a method that seems to guarantee a lower rate of fragmentation for polyps ≤ 9 mm [18,19].

If the polyp or fragment cannot enter the suctioning channel at all, they can be retrieved with specific devices. Graspers and forceps are an option but carry the risk of tissue maceration [17]. Baskets and retrieval nets [20,21] have the ability to collect multiple pieces without loss—in particular, retrieval nets seem to offer the largest capacity size and do not damage the sample [17]. These two devices have in common the important limitation of requiring the removal of the endoscope or, when they are held at a distance to continue the examination, offering an incomplete inspection of the colon [17]. On the other hand, snare retrieval has many disadvantages (limitation to a single polyp, tissue damage, and potential loss of the polyp) [17], but may allow continuing the examination if the polyp is small (5–8 mm). In this case, the snare can be pulled back 3–4 cm from the colonoscope tip, and, since sheath diameter is usually smaller than the colonoscope channel (2.3 vs. 3.5–3.8 mm), there is no need for instrument retrieval (channel occlusion technique) [22]; if the polyp is larger, however, the examination continues with the ensnared polyp pulled forward from the colonoscope, again partially occupying the visual field—unless the large polyp is periodically dropped [17].

Another described method of retrieval is clipping the polyps (before or after resection) and linking the clip to a nylon line, thus allowing a multiple polyp extraction [23,24], but this method appears to be time and cost-consuming.

The variability in polyp retrieval techniques highlights the need for an accurate polyp size measurement. In daily practice, endoscopists measure polyp size by visual estimation, leading to inaccuracy and interobserver variability [25]. In this scenario, a new virtual scale function (SCALE EYE^©^; Fujifilm, Tokyo, Japan) was developed in order to help endoscopists correctly measure polyps by projecting a virtual scale of measurement during real-time endoscopy. This new technology seems to improve correct polyp measurement and lower the interobserver variability [26].

#### 3.1.3. Collection of the Specimen

Gauze or a nylon mesh can be placed between the suction nipple and tubing to collect the resected polyp; however, these methods, despite their low cost, have the disadvantage of the potential maceration of the specimen [17]. Independent plastic compartments with a removable filter [27] or multicompartment traps [28] guarantee an ease of collection: even small fragments of 1–2 mm can be collected by the internal grid without being damaged. Suction requires passing the instrument channel, creating the problems of infections and risk of misdiagnosis (co-suction). Depending on the size, the retrieval can be through aspiration or the use of an endoscopic retrieval device [29]. The most commonly used methods are represented in Figure 1.

If the polyp is suctioned but not collected, injecting water—with a syringe through the instrument port or with the water-jet—with simultaneous suctioning may facilitate the retrieval of the polyp and reduce the risk of fragmentation [15,17]. Another retrieval method by suction is the so-called “water-slider method”: after resection, the specimen is carefully suctioned into the instrumental channel, but only minimally. The suction button is not pressed fully to avoid causing the polyp to pass through the suction tube without water. The operator activates the foot switch for the water-jet system to add sufficient water to the area near the resected polyp. Finally, the suction button is fully pressed, drawing the polyp into the suction tube along with the water to ensure it is properly collected. This method showed a reduced colonic polyp fragmentation rate [15]. If even this approach is unsuccessful in retrieval, after the instrument is withdrawn, the polyp can be searched by inspecting the biopsy channel with a cleaning brush, the biopsy orifice, underneath the surface of the rubber cap covering the biopsy port or the suction trumpet valve, and, lastly, by suctioning water in an attempt to flush the polyp out once the instrument is reassembled [14,28]. Inadequate bowel preparation, right colon location, and a previous colorectal surgery are other factors involved in polyp retrieval failure [3,16]. Fragmentation of the specimen during the retrieval process can interfere with the assessment of vertical and horizontal margins of the polyp, leading to an inadequate histological analysis [30].

### 3.2. Recommendations According to Guidelines

European Society of Gastrointestinal Endoscopy (ESGE) guidelines on performance measures for lower GI endoscopy [31] and those on quality screening colonoscopy [32] indicate, respectively, a minimum standard value of 90% (with a target greater than 95%) for the retrieval of polyps > 5 mm and a 90% for all polyps. British guidelines recommend similar standards [33] (*). The same values are reported for polyps of all sizes by the 2002 The United States Multi-Society Task Force (USMSTF) guidelines on quality in the technical performance of a colonoscopy [34]; however, the latest American Gastroenterology Association (AGA) paper on the quality of a screening colonoscopy [35] does not explicitly include polyp retrieval among quality indicators.

The importance associated with polyp retrieval seems obvious, since all existing guidelines on post-polypectomy surveillance (ESGE [5], USMSTF [36], British Society of Gastroenterology (BSG) [37]) base surveillance timing on histopathological examination; it is therefore crucial to (more or less implicitly) advise that, currently, all resect polyps must be retrieved in order to manage an endoscopic follow-up.

Follow-up colonoscopies are currently recommended based on the polyps’ size, number, and histological features; while the first and the second item are easily assessed by the endoscopist, the latter necessarily requires a pathological evaluation, thus increasing costs and delaying diagnosis and surveillance indication.

### 3.3. Resect and Discard Strategy: An Old Kid on the Block

In the last fifteen years, with the spread of high-resolution colonoscopy and virtual chromoendoscopy technologies in Western countries, a novel approach to manage colorectal lesions emerged in the debate in order to lower the cost of care: the “resect and discard” strategy, RD, for diminutive polyps.

Diminutive polyps are frequent findings during endoscopic examination [38] and, at the same time, carry the lowest risk of advanced histology—high-grade dysplasia, villous features—(0.5%) [39] and risk of cancer (0–0.08%) [40].

In 2011, the American Society for Gastrointestinal Endoscopy (ASGE) Technology Committee, in its Preservation and Incorporation of Valuable endoscopic Innovations (PIVI) paper, first opened to the idea that diminutive polyps could be resected and discarded without the need for an histology evaluation if proximal to the sigmoid colon; the sine qua non condition for this strategy is that the endoscopist must provide a high confidence optical diagnosis (OD) that can guarantee at least a 90% agreement in the assignment of post-polypectomy surveillance intervals when compared to decisions based on pathology assessments of all identified polyps. RD offers many advantages. First, it dramatically cuts the costs of histopathological analyses (estimated USD 33 million to over USD 1 billion per year, only in the US) [41,42,43,44]. Second, it reduces the workload for both pathologists and endoscopists [38]. Third, it can immediately assign a follow-up time to the patient [44]. This strategy is cost-effective without negatively affecting the quality of diagnosis, since studies have demonstrated a high concordance (≥90%) in post-polypectomy surveillance intervals in respect to the conventional histology-based method, thus meeting the ASGE 90% agreement threshold [45,46].

Based on this evidence, European and American endoscopic societies included RD strategies in polypectomy guidelines for diminutive polyps [37,47,48]; in addition, the ESGE described the general steps required for gaining and maintaining competence in OD of diminutive polyps [49,50]. Moreover, recently, RD policy has been developed by researchers in several possible ways, moving from the sole OD strategy to other possible approaches, such as the location-based strategy [51], polyp-based strategy [52,53] or simplified OD [54]. Despite the enthusiasm over the RD policy, the majority of endoscopists do not seem to agree with this strategy: the fear of making the wrong diagnosis or assigning the wrong interval, and the threat of medico-legal consequences are the principal obstacles to the implementation of this strategy [55,56]. Patients’ acceptance is another important issue [57]. Moreover, concrete protocols of OD for an RD policy seem so comprehensive that, at the moment, training and monitoring for an RD approach appears difficult and infeasible, at least on a large scale [58].

Thus, it is not surprising that current European and American guidelines on post-polypectomy surveillance, despite being the benchmark on which many studies on RD are conducted, do not mention the RD approach at all, creating an internal contrast between recommendations [5,36]. Only British guidelines are in favor of a follow-up colonoscopy on an OD-basis; however, they do not include this possibility in their post-polypectomy surveillance algorithm [37].

### 3.4. Histology Report and Surveillance Programs—Do We Need a Change?

In light of the above, the retrieval and subsequent histologic characterization for all resected lesions remain the core of post-polypectomy surveillance, and the only alternative to this is the waste of time and costs, that is, the RD policy for diminutive rectosigmoidal polyps currently has no indication on surveillance guidelines.

However, several studies questioned the accuracy of a pathological diagnosis in diminutive polyps, because tissue fragmentation during retrieval or failure to sectioning may lead pathologists to a misdiagnosis in such cases [46,59,60]. In the scenario of diminutive polyp resected and fragmented during retrieval, the histology could have no real advantage over the RD approach, since the histology could lose part of its accuracy. Furthermore, sending every diminutive polyp resected to a pathologist for evaluation creates a significant amount of workload for the pathologist and additional costs which cannot be justified given the very low risk of advanced histology of these tiny lesions.

In addition, given the technological advancements we now have at our disposal in endoscopy units (i.e., high-definition, magnification techniques and artificial intelligence), it is time for surveillance intervals modalities to be revised in light of the cost-effectiveness of optical diagnosis.

## 4. Conclusions

Our review has brought to light evidence on an important issue concerning the retrieval of removed polyps or fragments. It is clear that our manuscript has many limitations, as it is a narrative review and not an original contribution on this topic.

From this procedure commonly performed every day in large numbers comes an important pathway, each with pathological anatomy and surveillance protocols. As the literature increases on resection techniques, we think that it may also increase in how polyps and lesions are retrieved and requires some updated large-scale studies. Some strategies allow lesions reasonably free of potential adenomatous or malignant evolution to not be removed. However, many procedures depend, to a large extent, on the histological outcome that is expected by the patient themself. We hope that prospective studies analyzing this point can be developed and lead to better evidence on future surveillance protocols.

## Figures and Tables

**Figure 1 jcm-14-05846-f001:**
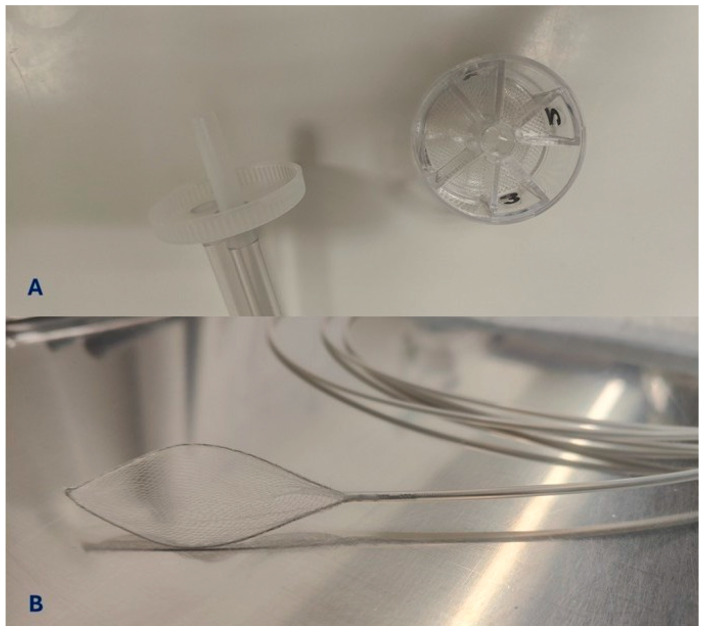
The most commonly used methods for polyps retrieval. (**A**). Polyp trap (SafeSnap 4+, © Olympus, Tokyo, Japan) with different compartments allowing for retrieval of small specimens by suction in the operative channel of the endoscope. (**B**). Oval loop net (Netis, © Meditalia, Genova, Italy), advanced through the scope used for the retrieval of large fragments not suitable for suction.

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
