# Peer review of "Resect and Retrieve Colorectal Polyps: Time for New Insights"

_jcm, 2025, doi:10.3390/jcm14165846_

Round 1

Reviewer 1 Report

Comments and Suggestions for Authors

Abstract:

Good abstract, very interesting, I think it whets the potential reader's appetite to read the full text.

The keywords are well chosen, suggestive for this manuscript.

On a scale of 1 to 10, I’ll give 10 points for this abstract.

Introduction:

The introduction is well done; the authors manage to convince the reader of the necessity and opportunity of this literature review. I believe that this chapter can remain in its current form.

On a scale of 1 to 10, I agree now 9 points for this introduction.

Methodology:

Unfortunately, there is no methodology, and for a literature review, even if it is of the narrative type, I consider the methodology to be mandatory, because it allows us to evaluate the scientific correctness of the study.

On a scale of 1 to 10, for a non-existing chapter, I must agree 0 (zero) points for methodology.

Results:

I would classify Chapter 2 (lines 71-159) as results. From this point of view, it is well done, sufficiently extensive, correctly supported with numerous bibliographical references. I think it could be kept as such only in terms of the text. However, this chapter has obvious shortcomings in the graphics, which are completely lacking, an aspect on which the authors deserve to reflect.

On a scale of 1 to 10, I agree 8 points for results.

Discussion:

All the other chapters of the manuscript would, in my opinion, fall under the discussion. I am referring to lines 160-237.

The discussion chapter should conclude with a brief summary of the strengths of the manuscript, followed by a presentation of the limitations of the study.

Graphical representations in this chapter would not be mandatory, but would add to the attractiveness of the manuscript.

In this situation, on a scale of 1 to 10, I agree only 6 points for discussion.

Conclusion:

There are no conclusions as such. I consider it mandatory that the authors write a short chapter of conclusions, which would preferably end with recommendations for future research directions.

On a scale of 1 to 10, for a non-existing chapter, I must agree 0 (zero) points for conclusions.

Bibliography/References:

61 bibliographic references correctly drafted and correctly cited in the text, from a strictly numerical point of view it could be acceptable. However, in the absence of a methodology, for a literature review, it is difficult to assess whether it is sufficient or not from the point of view of the bibliography.

On a scale of 1 to 10, I agree 7 points for the bibliography.

Figures/Tables:

Unfortunately, the manuscript contains nothing in the way of graphic representations.

On a scale of 1 to 10, for a non-existing chapter, I must agree 0 (zero) points.

Review Decision:

Reconsider after major revisions.

Author Response

Comments and Suggestions for Authors

Abstract:

Good abstract, very interesting, I think it whets the potential reader's appetite to read the full text.

The keywords are well chosen, suggestive for this manuscript.

On a scale of 1 to 10, I’ll give 10 points for this abstract.

We thank the Reviewer for this comment.

Introduction:

The introduction is well done; the authors manage to convince the reader of the necessity and opportunity of this literature review. I believe that this chapter can remain in its current form.

On a scale of 1 to 10, I agree now 9 points for this introduction.

We thank the Reviewer for this comment.

Methodology:

Unfortunately, there is no methodology, and for a literature review, even if it is of the narrative type, I consider the methodology to be mandatory, because it allows us to evaluate the scientific correctness of the study.

On a scale of 1 to 10, for a non-existing chapter, I must agree 0 (zero) points for methodology.

We thank the Reviewer for the suggestion. We added a paragraph explaining the methodology performed to search the literature. Since it was not a systematic review but a narrative one, we then decided how to organise the studies found in the results by thematic issue.

Results:

I would classify Chapter 2 (lines 71-159) as results. From this point of view, it is well done, sufficiently extensive, correctly supported with numerous bibliographical references. I think it could be kept as such only in terms of the text. However, this chapter has obvious shortcomings in the graphics, which are completely lacking, an aspect on which the authors deserve to reflect.

On a scale of 1 to 10, I agree 8 points for results.

Discussion:

All the other chapters of the manuscript would, in my opinion, fall under the discussion. I am referring to lines 160-237.

The discussion chapter should conclude with a brief summary of the strengths of the manuscript, followed by a presentation of the limitations of the study.

Graphical representations in this chapter would not be mandatory, but would add to the attractiveness of the manuscript.

In this situation, on a scale of 1 to 10, I agree only 6 points for discussion.

 We thank the Reviewer for this comment. We decided to include all the references as results; the discussion is part of the comments in each paragraph. As these are not our original results, we think the discussion can be developed paragraph by paragraph.

Conclusion:

There are no conclusions as such. I consider it mandatory that the authors write a short chapter of conclusions, which would preferably end with recommendations for future research directions.

On a scale of 1 to 10, for a non-existing chapter, I must agree 0 (zero) points for conclusions.

We thank the Reviewer for the suggestion. We added a paragraph including the conclusion.

Bibliography/References:

61 bibliographic references correctly drafted and correctly cited in the text, from a strictly numerical point of view it could be acceptable. However, in the absence of a methodology, for a literature review, it is difficult to assess whether it is sufficient or not from the point of view of the bibliography.

On a scale of 1 to 10, I agree 7 points for the bibliography.

We thank the Reviewer for this point and we explained our methodology in the dedicated paragraph.

Figures/Tables:

Unfortunately, the manuscript contains nothing in the way of graphic representations.

On a scale of 1 to 10, for a non-existing chapter, I must agree 0 (zero) points.

We thank the Reviewer for this comment. We added a figure representing the most commonly adopted way of retrieval.

Reviewer 2 Report

Comments and Suggestions for Authors

Dear All,

I was pleased to review the article “ Resect and retrieve colorectal polyps: time for new insights”.

This work- narrative review- is important, with an interesting and useful summary on small and diminutive colorectal polyps. As is known, most interval cancer are the result of a missed lesion (colorectal polyps) in previous colonoscopy. However, at the same time, the costs associated with sending all small and fragmented polyps to the pathologist, as well as repeating a colonoscopy at short intervals (surveillance programs), are unjustified.

The methodology used by the authors is appropriate for the purpose of the study. The methods of collecting polyps and the factors involved in polyp retrieval failure are explained.

In general, the manuscript may benefit from some revisions, as suggested below:

  • Line 16-17- I recommend replacing the term protocol with
  • should be used abbreviation for „resect and discard’’ (RD) starting with the first appearance in the text – line 40, 45...
  • NBI, BLI- line 58 and other abbreviations (AGA...) should be explained.
  • In section 5- Histology Report And Surveillance Programs Do We Need A Change? contains a kind of conclusion of what has been presented, namely sending every diminutive polyp resected to pathologist evaluation creates a significant amount of workload for pathologist and additional costs which cannot be justified given the very low risk of advanced histology of these tiny lesions. I recommend that those technological advancements it now dispose of in endoscopy units (listed as : high definition, magnification techniques and artificial intelligence) should be further detailed to support the idea of changing surveillance programs for patients with small and diminutive colorectal polyps.
  • The limitations of this narrative review also should be described.
  • The study have appropriate bibliographic references from recent years in accordance with this research- most of them being after the 2000s- but I suggest that  the bibliographical references should be written in the same way - see items 10, 60, 61. The item 23 - the volume is not mentioned.

Best regards,

Author Response

Dear All,

I was pleased to review the article “ Resect and retrieve colorectal polyps: time for new insights”.

This work- narrative review- is important, with an interesting and useful summary on small and diminutive colorectal polyps. As is known, most interval cancer are the result of a missed lesion (colorectal polyps) in previous colonoscopy. However, at the same time, the costs associated with sending all small and fragmented polyps to the pathologist, as well as repeating a colonoscopy at short intervals (surveillance programs), are unjustified.

The methodology used by the authors is appropriate for the purpose of the study. The methods of collecting polyps and the factors involved in polyp retrieval failure are explained.

In general, the manuscript may benefit from some revisions, as suggested below:

  • Line 16-17- I recommend replacing the term protocol with

We thank the Reviewer and changed the term “protocols” in “methods”.

  • should be used abbreviation for „resect and discard’’ (RD) starting with the first appearance in the text – line 40, 45...

We modified this point by adding the abbreviation from the first appearance.

  • NBI, BLI- line 58 and other abbreviations (AGA...) should be explained.

We added the explanations as required.

  • In section 5- Histology Report And Surveillance Programs Do We Need A Change? contains a kind of conclusion of what has been presented, namely sending every diminutive polyp resected to pathologist evaluation creates a significant amount of workload for pathologist and additional costs which cannot be justified given the very low risk of advanced histology of these tiny lesions. I recommend that those technological advancements it now dispose of in endoscopy units (listed as : high definition, magnification techniques and artificial intelligence) should be further detailed to support the idea of changing surveillance programs for patients with small and diminutive colorectal polyps.

We agree with the reviewer. We think that further studies on the common use of virtual chromoendoscopy and artificial intelligence (CADe or CADx systems) will bring to consistent evidence.

  • The limitations of this narrative review also should be described.

We thank the Reviewer for this comment and we added this point in the conclusion paragraph.

  • The study have appropriate bibliographic references from recent years in accordance with this research- most of them being after the 2000s- but I suggest that  the bibliographical references should be written in the same way - see items 10, 60, 61. The item 23 - the volume is not mentioned.

We thank the Reviewer for this comment and we modified the suggested references.

Round 2

Reviewer 1 Report

Comments and Suggestions for Authors

No further suggestions